# Circulating miRNAs as Noninvasive Biomarkers for PDAC Diagnosis and Prognosis in Mexico

**DOI:** 10.3390/ijms242015193

**Published:** 2023-10-14

**Authors:** Lissuly Guadalupe Álvarez-Hilario, Eric Genaro Salmerón-Bárcenas, Pedro Antonio Ávila-López, Georgina Hernández-Montes, Elena Aréchaga-Ocampo, Roberto Herrera-Goepfert, Jorge Albores-Saavedra, María del Carmen Manzano-Robleda, Héctor Iván Saldívar-Cerón, Sandra Paola Martínez-Frías, María Del Rocío Thompson-Bonilla, Miguel Vargas, Rosaura Hernández-Rivas

**Affiliations:** 1Departamento de Biomedicina Molecular, Centro de Investigación y de Estudios Avanzados del Instituto Politécnico Nacional, Ciudad de Mexico C.P. 07360, Mexico; lissulyalvarez@gmail.com (L.G.Á.-H.); eric.salmeron@cinvestav.mx (E.G.S.-B.); pedroavila@cinvestav.mx (P.A.Á.-L.); hector.saldivar@cinvestav.mx (H.I.S.-C.); mavargas@cinvestav.mx (M.V.); 2Coordinación de la Investigación Científica, Red de Apoyo a la Investigación, Universidad Nacional Autónoma de México, Ciudad Universitaria, Ciudad de Mexico C.P. 14080, Mexico; yinna@cic.unam.mx; 3Departamento de Ciencias Naturales, Universidad Autónoma Metropolitana, Unidad Cuajimalpa, Ciudad de Mexico C.P. 05300, Mexico; e.arechaga@dcniuamc.com; 4Departamento de Patología, Instituto Nacional de Cancerología, Ciudad de Mexico C.P. 14080, Mexico; rhgoepfert@gmail.com; 5Departamento de Patología, Medica Sur Clínica y Fundación, Ciudad de Mexico C.P. 14050, Mexico; dralboresjorge@gmail.com; 6Servicio de Endoscopía Gastrointestinal, Instituto Nacional de Cancerología, Ciudad de Mexico C.P. 14080, Mexico; macmanzano@gmail.com; 7Departamento de Infectología, Instituto Nacional de Ciencias Médicas y Nutrición Salvador Zubirán (INCMNSZ), Avenida Vasco de Quiroga No.15, Colonia Belisario Domínguez Sección XVI, Ciudad de Mexico C.P. 14080, Mexico; 8Laboratorio de Medicina Genómica, Hospital Regional 1° de Octubre, Ciudad de Mexico C.P. 07300, Mexico; rociothompson@yahoo.com.mx

**Keywords:** PDAC, miRNAs, microarrays, biopsies, plasma

## Abstract

Among malignant neoplasms, pancreatic ductal adenocarcinoma (PDAC) has one of the highest fatality rates due to its late detection. Therefore, it is essential to discover a noninvasive, early, specific, and sensitive diagnostic method. MicroRNAs (miRNAs) are attractive biomarkers because they are accessible, highly specific, and sensitive. It is crucial to find miRNAs that could be used as possible biomarkers because PDAC is the eighth most common cause of cancer death in Mexico. With the help of microRNA microarrays, differentially expressed miRNAs (DEmiRNAs) were found in PDAC tissues. The presence of these DEmiRNAs in the plasma of Mexican patients with PDAC was determined using RT-qPCR. Receiver operating characteristic curve analysis was performed to determine the diagnostic capacity of these DEmiRNAs. Gene Expression Omnibus datasets (GEO) were employed to verify our results. The Prisma V8 statistical analysis program was used. Four DEmiRNAs in plasma from PDAC patients and microarray tissues were found. Serum samples from patients with PDAC were used to validate their overexpression in GEO databases. We discovered a new panel of the two miRNAs miR-222-3p and miR-221-3p that could be used to diagnose PDAC, and when miR-221-3p and miR-222-3p were overexpressed, survival rates decreased. Therefore, miR-222-3p and miR-221-3p might be employed as noninvasive indicators for the diagnosis and survival of PDAC in Mexican patients.

## 1. Introduction

Pancreatic ductal adenocarcinoma (PDAC) has one of the highest mortality rates among malignant tumors worldwide and is responsible for 3.2% of new cancer cases and 7.2% of all cancer-related deaths. PDAC is the fourth-leading cause of cancer-related death in the United States [1] and the eighth-leading cause of cancer-related death in Mexico (WHO. https://gco.iarc.fr/ accessed on 5 April 2023); it is predicted to become the second-leading cause of cancer-related death in the United States by 2030 [2].

PDAC is characterized by resistance to chemotherapy [3], diagnosis at an advanced stage, and early dissemination of tumor-derived cells into the bloodstream [4]. The optimal treatment for patients with PDAC is surgical resection, but only patients with localized tumors are eligible for this treatment, and most patients are diagnosed at an advanced stage [5]. These factors lead to high mortality and a 5-year survival rate of only 10%; more than 80% of patients diagnosed with PDAC die from the disease [1]. The lack of effective screening tools to detect asymptomatic premalignant cases or early-stage PDAC contributes to its high mortality. The differential diagnosis remains challenging, and ideally, invasive techniques would not be used for this diagnosis. In this context, the level of CA19-9 is used to evaluate disease progression in patients with PDAC. Still, the carbohydrate antigen sialyl Lewis (CA19-9) level also increases in nonmalignant pancreatic conditions, but it is not a recommended biomarker for PDAC screening [6] because it can produce false-negative results [6]. Carcinoembryonic antigen (CEA) is another tumor marker, but it lacks high specificity and sensitivity [6]. Therefore, although the histological studies and imaging techniques on which PDAC diagnosis still relies have improved, it is imperative to identify more sensitive, marker-specific, and noninvasive procedures.

MicroRNAs (miRNAs) have emerged as promising cancer biomarkers. miRNAs are endogenous, highly conserved, short noncoding RNAs that are ~23 nt in length; miRNAs negatively influence posttranscriptional gene expression to regulate development, normal physiology, and disease [7]. Altered miRNA expression has been reported in almost all human cancers [8]. 

Furthermore, miRNAs play important roles in cancer biology by promoting tumor growth, invasion, angiogenesis, and immune evasion by regulating target gene expression [9]. MiRNAs have significant potential to be used as biomarkers for diagnosis, prognosis, and the assessment of risk factors, as well as to monitor the progression and stage of the disease. This is because the genesis and metastasis of cancer are linked to altered miRNA expression. Since miRNAs are stable molecules, their measurement does not require large amounts of samples. In addition, the altered expression of miRNAs starts in the early stages of the disease, making them beneficial for early cancer diagnosis [10]. Additionally, they can be measured in fresh tissue biopsies and even in paraffin-embedded tissue [11], as well as in some biological fluids, such as serum and plasma, which offers a less invasive method for their monitoring [12]. Therefore, many studies have evaluated the potential of circulating miRNAs (circulating or exosome-bound miRNAs) as noninvasive biomarkers for specific diseases, including PDAC [13,14]. Independent studies have identified several miRNAs in the serum or plasma of patients with PDAC [15,16,17,18,19,20,21,22]. However, the genetic heterogeneity in each population affects miRNA expression and/or function in individuals with different ethnic origins. In addition, not all studies have compared serum and plasma miRNA levels with corresponding changes in pancreatic tumor tissue. 

In this study, we identified DEmiRNAs in tissues from patients with PDAC who reside in Mexico and compared them to those from healthy individuals to identify miRNAs overexpressed in the plasma of patients with PDAC to determine a miRNA signature in Mexican patients with this cancer. We found that overexpression of miR-222-3p and miR-221-3p in plasma showed high diagnostic sensitivity and specificity, indicating their value as novel biomarkers for the noninvasive diagnosis of PDAC. miR-221-3p and miR-222-3p were significantly associated with poor outcomes in patients with PDAC, highlighting their roles as potential prognostic factors.

## 2. Results

### 2.1. Identification of DEmiRNAs in PDAC

To identify DEmiRNAs in patients with PDAC who live in Mexico, we used microarrays to measure miRNA expression in 12 of 25 PDAC tumors and 3 of 13 samples of non-neoplastic pancreatic tissue from the first cohort of samples (Figure 1 and Table 1).

Previously, we confirmed the presence of tumor cells in all biopsy samples of PDAC tumors and control tissues by cytokeratin 7 staining and histological analyses (Figure 2A,B). As expected, in the nonneoplastic pancreatic samples, the ductal cells presented a normal morphology with acini in the form of lobules and ducts with a simple cubic epithelium, and no neoplastic cells were observed (Figure 2A). In contrast, a low degree of cell differentiation was observed in the tissues of patients with PDAC; the ducts were no longer present, and there was a greater amount of stroma (Figure 2B). To isolate the DEmiRNAs in the neoplastic cells, excluding the miRNAs expressed in the stroma, neoplastic cells were isolated from control and PDAC tissues by macrodissection (Figure 2C). Then, total RNA was isolated from the macrodissected samples, and miRNA expression levels were analyzed by microarrays. Principal component analysis (PCA) of the miRNA expression profiles of tissues from PDAC patients and control subjects showed that the neoplastic and nonneoplastic samples were properly classified (Figure 2D). Seventeen DEmiRNAs were identified between patients with PDAC and controls (Figure 2E and Table 2).

Of the 17 DEmiRNAs, 16 were overexpressed, and only miR-148–3p was downregulated (Figure 2E). The expression profile of the 17 DEmiRNAs was clearly able to distinguish the PDAC samples from the non-neoplastic samples, as shown in the heatmap (Figure 2F).

### 2.2. Systematic Validation of DEmiRNAs in PDAC

To validate the overexpression of the 12 miRNAs with the highest *p*-values, RT-qPCR experiments were performed on pooled biopsy samples from 10 control participants and 13 patients with PDAC (the rest of the samples from the first cohort) (Figure 1 and Table 1). The results confirmed that all 12 miRNAs were overexpressed (Figure 2G). To corroborate the signature of the 12 overexpressed miRNAs in PDAC, the expression of those miRNAs was evaluated in an independent second cohort of patients and controls. Total RNA was obtained from tumor tissues of 15 patients with PDAC and nonneoplastic pancreatic tissue from 15 individual controls, and the expression of 12 DEmiRNAs was analyzed by RT-qPCR assays (Figure 1 and Table 1). In this second patient cohort, all 12 miRNAs were found to be overexpressed, but the differences were statistically significant for only 10 miRNAs (miR-222–3p, miR-210–3p, miR-10b-5p, miR-203a-3p, miR-10a-5p, miR-345–5p, miR-155–5p, miR-100–5p, miR-221–3p, and miR-150–5p) (Figure 2H). All these data suggest that we discovered a signature of 10 miRNAs that can be used to identify PDAC tumors.

### 2.3. Identification of miRNA Signatures in the Plasma of Patients with PDAC

Based on these findings, the miRNA signatures in the plasma of patients with PDAC were analyzed to determine the potential role of these miRNAs as diagnostic biomarkers. Thus, we determined which of the 10 miRNAs identified in tumor tissues from patients with PDAC could be detected in the bloodstream of patients with PDAC. miRNAs were isolated from plasma samples of 46 patients with PDAC and 20 nonneoplastic controls (Figure 1 and Table 1), and the miRNAs expression levels were evaluated by RT-qPCR. The results indicated that 4 of 10 miRNAs (miR-222–3p, miR-345-5p, miR-100–5p and miR-221-3p) were significantly overexpressed in the plasma of patients with PDAC (Figure 3A). Then, we corroborated the expression of these four DEmiRNAs in PDAC patient miRNA data obtained from two GEO datasets. The first dataset analyzed, GSE106817, contained miRNA expression data from the sera of 115 patients with PDAC and 2759 controls (Figure 3B) [23]. The second dataset, GSE112264, contained data from 50 PDAC patient sera and 41 control sera (Figure 3B) [24].

These analyses showed that three of the four DEmiRNAs (miR-222-3p, miR-100-5p and miR-221-3p) were also significantly and more highly expressed in a large cohort of PDAC patient serum samples (Figure 3C,D). Unexpectedly, we found that miR-345-5p was not overexpressed in the serum sample data in the databases, probably because the overexpression of this miRNA is specific to patients with PDAC who live in Mexico.

All these data indicated that miR-222-3p, miR-345-5p, miR-100-5p, and miR-221-3p were overexpressed in the plasma of patients with PDAC residing in Mexico, highlighting their potential use as diagnostic markers.

### 2.4. Serum miRNA Signatures with Potential Value in the Diagnosis of PDAC

To evaluate the diagnostic value of the four DEmiRNAs (miR-222-3p, miR-345-5p, miR-100-5p, and miR-221-3p) identified in the plasma of patients with PDAC, receiver operating characteristic (ROC) curve analyses were performed, and the area under the curve (AUC) was calculated together with the sensitivity and specificity for each miRNA identified in this study. An AUC equal to 0.5 generally indicates that there is no distinction between individuals with and without the disease or condition based on the test, whereas an AUC of 0.7 to 0.8 is deemed acceptable [25]. Our findings showed that the fourth miRNAs (miR-22-3p, miR-345-5p, miR-100-5p, and miR-221-3p) had an AUC > 0.7 (Figure 4A). Even though miR-222-3p and miR-221-3p had the best AUCs of 0.7266 and 0.7384, respectively, miR-222-3p had a higher sensitivity of 73.91% than miR-221-3p at 63.4%. These findings imply that miR-222-3p may be the most accurate biomarker among the four miRNAs discovered.

Then, to improve the specificity and sensitivity of the miR-222-3p, miR-345-5p, miR-100-5p, and miR-221-3p, we assessed a series of combinations by comparing the expression levels of two miRNAs (miR-222-3p/miR-345-5p; miR-222-3p/miR-100-5p; miR-222-3p/miR-221-3p; miR-345-5p/miR-100-5p; miR-345-5p/miR-221-3p; miR-100-5p/miR-221-3p), or three miRNAs (miR-222–3p/miR-345-5p/miR-100-5p; miR-222-3p/miR-345-5p/miR-221-3p; miR-100-5p/miR-221-3p/miR-222-3p; miR-100-5p/miR-221-3p/miR-345-5p) and the four miRNA combinations (miR-222-3p/miR-345-5p/miR-100-5p/miR-2221-3p), resulting in the generation of 11 tables, one table for each combination of miRNAs examined. Using the default parameters, ROC curves were generated using these data. Finally, the AUC was calculated for each combination using a 95% CI, and an AUC of less than 7 with a value of 0.05 was considered sufficient. Finally, the AUC was calculated for each combination with a 95% CI, and an AUC >7 with a value of *p* < 0.05 was considered sufficient [23]. Using the Youden index method, the best cutoff values, sensitivity, and specificity for each miRNA combination were identified.

We discovered that, while considering the AUC, sensitivity, and specificity, the combination miR-222-3p/miR-221-3p exhibited the best balance out of the three combinations (miR-222-3p/miR-345-5p, miR-222-3p/miR-100-5p and miR-345-5p/miR-100-5p). Once more, miR-222-3p/miR-221-3p produced a better AUC, sensitivity and specificity (Figure 4B) than the combinations of three or four miRNAs (Figure 4C,D). Therefore, we propose that overexpression of miR-222-3p and miR-221-3p could be considered a good noninvasive biomarker for PDAC diagnosis.

Interestingly, a reanalysis of the RT-qPCR data obtained from biopsies and plasma of PDAC patients who lived in Mexico indicated that miR-222-3p was expressed in stage IIA and stage IV PDAC (Appendix A). miRNA-221-3p was transcribed from stages II to IV (Appendix A). Therefore, the combination of miR-222-3p/miR-221-3p could be used to diagnose PDAC from early (IIA) to late stages (IV) of PDAC.

### 2.5. miR-221-3p and miR-222-3p as Biomarkers of Poor Survival in Patients with PDAC

We used Kaplan-Meier curve analysis with the TCGA database to examine the associations between the expression levels of the four DEmiRNAs (miR-221-3p, miR-222-3p, miR-345-5p and miR-100-5p) identified in serum/plasma and the overall survival (OS) rates of patients with PDAC. Our results suggest that only two of the four miRNAs (miR-222-3p and miR-221-3p) are related to a lower survival rate (Figure 5A) because only these two miRNAs had a *p*-value < 0.05. To determine whether the combination of both miRNAs increased their predictive value, we performed miRNA combinations between miR-222-3p and miR-221-3p (Figure 5B). Interestingly, when we compared the results obtained with miR-222-3p and miR-221-3p alone or combined, we found that miR-221-3p alone had a poorer survival than the miR-222-3p/miR-221-3p combination (Figure 5B). As a result, miR-222-3p and miR-221-3p, either individually or together, are associated with poor survival in patients with PDAC.

### 2.6. Target Gene Prediction and Functional Analysis of Four DEmiRNAs in PDAC

To understand the biological functions of four DEmiRNAs in patients with PDAC, we conducted a bioinformatic analysis to predict the putative target genes of four DEmiRNAs (miR-222–3p, miR-345-5p, miR-100–5p, and miR-221–3p) using the miRPathDB online analysis tool, and 21,872 putative mRNA targets were identified (Appendix A). To determine which of these mRNAs truly exhibited downregulated expression in PDAC tumor samples, we used two GEO databases containing gene expression profile datasets of PDAC tumor tissues: GSE62452 [26] and GSE28735 [27] (Figure 6A). The results revealed 277 differentially expressed genes (DEGs) in the GSE62452 database and 390 DEGs in the GSE28735 database (Appendix A). Of these, 173 overexpressed and 98 downregulated genes were common to both datasets (Figure 6B). Importantly, 47 of the 98 downregulated genes were identified as target genes of one of the four DEmiRNAs (Appendix A). To elucidate the biological functions of the 47 downregulated genes, a KEGG pathway enrichment analysis was performed. The results showed that the genes were mainly enriched in pathways associated with maturity-onset diabetes of the young, the ErbB signaling pathway, the FoxO signaling pathway, fatty acid degradation, the JAK-STAT signaling pathway, glycolysis/gluconeogenesis, and pancreatic secretion (Figure 6C). Finally, these genes were also involved in biological processes such as negative regulation of epidermal cell differentiation, regulation of small molecule metabolic processes, fatty acid oxidation, regulation of steroid metabolic processes, and positive regulation of carbohydrate metabolic processes (Figure 6D). These results suggested that metabolic adaptation might facilitate pancreatic tumor growth proteolysis and cell junction organization, positive regulation of epithelial cell and keratinocyte proliferation, cellular invasion, and migration (Figure 6D). All these results indicate that the set of miRNAs whose expression is dysregulated in tumor tissues and plasma in patients with PDAC might regulate pathways and biological processes related to the tumorigenesis and progression of PDAC.

### 2.7. Putative Target Genes of Four DEmiRNAs in PDAC Tumors Represent Prognostic Factors in Patients with PDAC

The prognostic value of the 47 putative target genes of the four DEmiRNAs in patients with PDAC was assessed by Kaplan–Meier curve analysis of data from the TCGA databases. We found that 11 out of the 47 genes that were expressed at low levels (*C5, CTNND2, EPHX2, ERO1B, F8, KIAA1324, NRCAM, NUCB2, PAK3, SLC7A2, and TTR*) were associated with poor OS (Appendix A). Therefore, it is conceivable that low expression of these 11 genes might have an impact on the prognosis of PDAC.

Using the TCGA dataset and the GEPIA database, we ascertained whether those 11 targets of the four DEmiRNAs were downregulated at the RNA level in patients with PDAC compared to controls. As expected, the four DEmiRNAs had a downregulating effect on 9 of their 11 target mRNAs (*C5, CTNND2, EPHX2, ERO1B, KIAA1324, NRCAM, NUCB2, PAK3,* and *TTR*) (Appendix A).

To test the prognostic value of the low expression of the proteins encoded by these 11 genes, immunohistochemical staining data obtained from the Human Protein Atlas database of PDAC patient tissues were analyzed. We found that the proteins encoded by 4/11 genes, namely, EPHX2, EROIB, KIAA1324, and NRCAM, were expressed at lower levels in PDAC tissues than in normal pancreatic tissues (Appendix A). These data suggest that these four proteins may serve as prognostic markers for PDAC.

## 3. Discussion

In Mexico, pancreatic cancer, specifically PDAC, is the eighth-leading cause of cancer-related death (WHO. https://gco.iarc.fr/ accessed on 5 April 2023). However, no study has been performed to identify DEmiRNAs in Mexican patients with PDAC that could be used to diagnose this type of cancer. For this reason, in this work, we first proceeded to identify DEmiRNAs in biopsies of patients with PDAC to determine which of these miRNAs could be present in the plasma of patients with PDAC. Importantly, we identified four DEmiRNAs (miR-222-3p, miR-345-5p, miR-100-5p, and miR-221-3p) that were overexpressed both in biopsies and in plasma from patients who reside in Mexico with this type of cancer. These four DEmiRNAs have already been identified in biopsies of patients with PDAC in different studies [28,29,30,31,32,33] and have been detected in the serum/plasma of patients with PDAC from the US, China, Japan, and Germany [32,33,34,35] and now, for the first time, in the Mestizo-Mexican patients analyzed in this study. These results indicate that the expression of these miRNAs is conserved among different populations, so these miRNAs could be potential biomarkers for the diagnosis of PDAC.

In this work, we focused on several characteristics that make a biomarker useful for the diagnosis of PDAC, such as the AUC, specificity, sensitivity, and *p*-value. Considering all of these parameters, our results show that the miR-222-3p/miR-221-3p combination exhibited a better balance of specificity, sensitivity, and *p*-value. The choice of these two miRNAs as promising molecules for diagnosis is supported by Lee C., who discovered that miR-221-3p and miR-222-3p were considerably increased in pancreatic cancer tissue. Additionally, the reliability of miR-222-3p and miR-221-3p as biomarkers for the diagnosis of several diseases has also been demonstrated more recently [36]. Therefore, all the published information and our data support the use of miR-222-3p and miR-221-3p as noninvasive biomarkers for the diagnosis of PDAC. However, we recognize that more Mexican PDAC cases are needed to support our findings.

Additionally, we discovered that miR-221-3p and miR-222-3p alone might be employed as PDAC indicators. Our findings have been supported by Li F’s 2018 studies, which showed that miR-221-3p is independently overexpressed in pancreatic cancer tissues and blood and may serve as a biomarker for the disease [37]. Furthermore, it was found that the expression level of miR-221-3p correlates with distant metastases and TNM stage [37]. For miR-222-3p, Dittmare found in 2021 that this miRNA, together with miR-34a-5p and miR-130-5p, is abundant in the plasma of patients with PDAC in stage II and may be used for the early diagnosis of pancreatic cancer [16]. Currently, only three distinct panels of two miRNAs have been identified as potential diagnostic markers for this illness [34,38,39]. Therefore, miR-221-3p and miR-222-3p have been proposed for the first time as diagnostic biomarkers from the blood of patients with PDAC. Several studies have analyzed the combination of miRNAs with the CA-19–9 antigen, which significantly increases the precision of diagnostic tests [34]. Unfortunately, we did not have data about the CA19-9 levels of control individuals in this study; therefore, this combination, which might improve upon the sensitivity and specificity of the two miRNAs identified, was not tested.

MiRNAs can also serve as prognostic biomarkers. In this study, we discovered that miR-222-3p and miR-221-3p could be useful as biomarkers for patients with PDAC who have a poor prognosis. Therefore, miR-222-3p and miR-221-3p could potentially act as biomarkers of poor survival in patients with PDAC in addition to being employed as noninvasive diagnostic biomarkers. Recently, Jialing Song published that miR-221/miR-222 has potential value as a prognostic biomarker in several cancers, including pancreatic cancer [36], strengthening the idea that miR-222-3p and miR-221-3p could be used as novel noninvasive prognostic biomarkers in PDAC patients who live in Mexico.

Metabolic reprogramming is one of the hallmarks of cancer cells and is characterized by an increase in glucose uptake and aerobic glycolysis [40]. This reprogramming of the glycolytic pathway occurs because cancerous tissue consumes much more glucose than normal tissue, and much of this glucose is converted to lactate in the presence of oxygen, which promotes an acidic and immunosuppressive tumor microenvironment. Acidification of the extracellular matrix activates matrix metalloproteases, which leads to degradation of the matrix and subsequent invasion of cancer cells. Aerobic glycolysis suppresses oxidative metabolism, induces anoikis resistance, and promotes cell migration and the tumor microenvironment. Pancreatic cancer cells also reprogram amino acid metabolism by increasing the transport of glutamine since glutamine metabolism allows cancer cells to support their growth and proliferation through the production of energy and the biosynthesis of proteins, lipids, and nucleic acids and contributes to redox homeostasis and cell signaling. Amino acids are an important source of nitrogen and carbon atoms that are used in the biosynthesis of macromolecules [41,42]. Pancreatic cancer cells reactivate lipogenesis, which is necessary for tumorigenesis, cancer progression, and even the aggressive behavior of this cancer. Consistent with these data, we found that the main pathways that are dysregulated by these four DEmiRNAs through their target mRNAs correspond to the oxidation of fatty acids, glucose homeostasis, catabolic processes of nitrogenous compounds, and the transport of amino acids, all of which are metabolic processes that are known to be reprogrammed to allow cancer cells to survive and proliferate [43,44]. Therefore, our results indicate that the four DEmiRNAs identified seem to be involved in regulating the expression of genes involved in metabolic reprogramming and DAC pathobiology.

Another mechanism that is dysregulated by these four DEmiRNAs is related to various signal transduction pathways. This can be explained by the fact that the common signaling molecules involved in metabolic pathways are phosphatidylinositol 3-kinase (PI3K), protein kinase B (known as AKT), adenosine monophosphate-activated protein kinase (AMPK), and mammalian target of rapamycin (mTOR) [45,46]. Consequently, our results suggest that the four DEmiRNAs might target molecules that dysregulate pathways associated with energy production and biosynthesis, thus reinforcing the metabolic reprogramming of pancreatic cancer cells [47,48].

Finally, we were able to establish that the low expression levels of 11 possible target genes of the four DEmiRNAs correlated with poor OS, so it is conceivable that some of these target genes could have an impact on the prognosis of PDAC. This is supported by the fact that several of the mRNAs identified in this work have been reported to be markers of poor prognosis. For example, it has been reported that low expression levels of KIAA1324 are associated with poor prognosis in patients with gastric cancer [49]. NUCB2 has been associated with poor prognosis in breast cancer, prostate cancer, endometrial carcinoma, and bladder cancer [50], while low expression levels of *SLC7A2* in ovarian cancer were associated with a positive prognosis. Similar results have been reported for TTR PDAC prognosis [51,52]. These results suggest that the expression levels of these genes may be additional prognostic markers. We also found that low protein expression levels of EPHX2, EROiB, KIAA1324, and NRCAM in tissue samples from patients with PDAC correlated with poor survival. Together, these results suggest that at the histological level, these four proteins could serve as prognostic biomarkers for patients with PDAC.

## 4. Materials and Methods

### 4.1. The Tissue and Plasma Samples

The Department of Pathology at Instituto Nacional de Cancerología (INCan) supplied formalin-fixed and paraffin-embedded tissues (FFPE) from 25 PDAC surgical specimens and 13 noncancerous pancreatic tissue samples close to the tumor from postmortem examinations (Table 1 and Appendix A).

Individuals diagnosed with PDAC without treatment, who had not suffered from pancreatitis or diabetes mellitus, and who did not have a tumor other than PDAC were included in the study. The Department of Pathology and the Biobank of INCan in Mexico City, Mexico, provided plasma samples from 66 people (46 patients with PDAC and 20 controls), as shown in Table 1 and Appendix A. Patients with a diagnosis of PDAC who had not previously received treatment (chemo- and/or radiation), did not have any other pancreatic disease, and agreed to participate in the study were eligible for the PDAC plasma set. The plasma of control subjects was collected from those with no history of cancer, diabetes, or pancreatitis after signing an informed consent form.

### 4.2. Ethics Approval and Consent to Participate

The study was carried out in accordance with all ethical standards regulating the use of human volunteers, as well as the Helsinki Declaration’s guidelines. For the biopsies and plasma samples, the Research and Ethics Committee of INCan Mexico City gave its approval (approval number: (INCAN/CI/1110/17 and CEI/1401/19). This study was also approved by the Committee of Bioethics of Human Health (COBISH) of CINVESTAV (035/2016). After signing the informed consent form, the samples were taken.

### 4.3. Tissue Macrodissection and Total RNA Extraction

FFPE tissue sections were obtained from surgically resected tumors (5–10 sections each, 10 μm thick). To isolate the total areas of the tumor and exclude stromal cells, tissue macrodissection was performed. First, tissue sections were prepared for needle macrodissection using an Olympus stereomicroscope. A representative example of macrodissected areas (tumor and histologically normal pancreatic tissues) is shown in Figure 2A,B. Total RNA was isolated from macrodissected samples using the PureLink FFPE Total RNA Isolation kit (Invitrogen, Waltham, MA, USA) according to the manufacturer’s instructions. The integrity of the isolated total RNA was analyzed using a 1.2% agarose gel. Purified total RNA was quantified using a Nanodrop ND1000 (NanoDrop Technologies, Thermo Fisher Scientific, Waltham, MA, USA).

### 4.4. Plasma miRNA Isolation and cDNA Synthesis

For the evaluation of circulating miRNAs, whole blood samples were collected from healthy individuals and PDAC patients in EDTA-containing tubes (BD Vacutainer; Becton Dickinson and Company, Franklin Lakes, NJ, USA) and processed as described by the manufacturer. Total RNA, primarily miRNAs, was obtained by using a miRNeasy serum/plasma kit (Qiagen, Hongkong, China) according to the manufacturer’s instructions with some modifications. Briefly, 600 µL of plasma was used to purify miRNAs from each sample. We used miR-39-3p from *Caenorhabditis elegans* as a spike-in control, which was added before performing chloroform extraction [53]. From each sample, the RNA was eluted in 15 µL of RNase-free water. The quantification of RNA was performed by using a NanoDrop ND-2000 (Thermo Fisher Scientific, Waltham, MA, USA). One hundred nanograms of RNA was used for cDNA synthesis for each sample. miRNAs were reverse-transcribed according to Bush PK’s protocol [54].

### 4.5. miRNA Microarray Assays

A GeneChip microRNA 4.0 array (Thermo Fisher Scientific) containing probes representing 2588 miRNAs (miRBase release 21; www.mirbase.org accessed on 24 August 2018) was performed according to the protocol described by Felix F. Tainara [55]. The data analysis was executed using Expression Console software v.4.0.2 (Affymetrix, Santa Clara, CA, USA) for data summarization, normalization, and quality control. To identify significantly deregulated miRNAs, the following parameters were used in each tumor compared to normal tissues: fold change (FC) ≥ 2, *p*-value < 0.0001 and a false discovery rate (FDR) < 0.05 were used to compare each tumor to normal tissues.

### 4.6. miRNA Differential Expression Analysis

Background correction and normalization of microarray data were performed using RMA [56]. Statistical analysis was performed using limma, a parametric approach using linear models and empirical Bayes [57]. Heatmaps were plotted using R functions (https://cran.r-project.org/ accessed on 30 August 2018). DEmiRNAs were annotated with miRNA-4_0-st v1.annotations.20160922.csv, a database provided by Affymetrix.

### 4.7. miRNA Expression by RT-qPCR

miRNAs were reverse-transcribed according to Busk PK’s protocol (54). Briefly, cDNA synthesis was performed with 100 ng of total RNA and 1 µL (10X) of Escherichia buffer, coli poly(A) polymerase, 1 µL (1 mM) ATP, 1 µL (10 µM) RT primer 5′-CAGGTCCAGTTTTTTTTTTTTTTTTTVN, 1 µL (10 mM) dNTP mix, 0.5 µL M-MulV reverse transcriptase (200 U/µL), and 0.2 µL *E. coli* poly(A) (5000 U/mL). Real-time PCR assays were performed on a Step One real-time system (Applied Biosystems). Syber green (Maxima SYBR Green/ROX qPCR Master Mix, 2X, Thermo Fisher Scientific) and specific primers (Table 3) were used to quantify each miRNA in the real-time PCR assay. The primers for each miRNA were designed using miRprimer V2.0 software (https://sourceforge.net/projects/mirprimer/ accessed on 17 September 2018) as described by Busk PK, 2014 [58]. miRNAs relative levels were determined by 2^−ΔCt^ method [58] and normalized using the RNU6 or miR-39-3p controls as housekeeping miRNAs for tissue and plasma samples, respectively.

### 4.8. Bioinformatics Analysis of miRNAs and Gene Expression

To assess the expression of the four DEmiRNAs in the data from patients with PDAC, two Gene Expression Omnibus (GEO) datasets were employed: GSE106817 [23] and GSE112264 [24], whereas the GSE62452 and GSE28735 datasets were used to determine the expression of putative target genes of these DEmiRNAs using GEO2R software v.4.2.2 (https://www.ncbi.nlm.nih.gov/geo/geo2r/ accessed on 27 November 2020) [59]. Samples were divided into PDAC and controls for each dataset, and expression analysis was performed using default parameters between these two groups. For each dataset, the entire results table was obtained, generating a total of four tables. We used GEO2R software v.4.2.2 to search for each miRNA’s ID probe and download the expression data for each miRNA by sample for miRNA expression analysis. Differences between the control and PDAC groups were determined using the Mann-Whitney test in GraphPad Prism V8.0 software (GraphPad Software, La Jolla, CA, USA), * *p*-value ˂ 0.05, ** *p*-value ˂ 0.01 and *** *p*-value ˂ 0. 001. For the expression analysis of putative target genes of these DEmiRNAs, an FC ≥ 1 and a *p*-value < 0.05 were used as cutoffs from the data downloaded in GEO2R v.4.2.2.

DEmRNA expression was validated in data from patients with PDAC in The Cancer Genome Atlas (TCGA) and Genotype-Tissue Expression (GTEx) datasets using the Gene Expression Profiling Interactive Analysis (GEPIA) database (http://gepia.cancer-pku.cn/ accessed on 24 November 2020) [60]. DEmRNA expression was compared between PDAC and normal tissue samples using default parameters, and the differences were determined using one-way ANOVA. * *p*-value ˂ 0.05. Validation was also performed in patients with PDAC using the Human Protein Atlas Version 23 database (https://www.proteinatlas.org/ accessed on 24 November 2020) [61] (https://www.proteinatlas.org/ENSG00000120915-EPHX2/tissue/pancreas#img; https://www.proteinatlas.org/ENSG00000120915-EPHX2/pathology/pancreatic+cancer#img; https://www.proteinatlas.org/ENSG00000116299-ELAPOR1/tissue/pancreas#img; https://www.proteinatlas.org/ENSG00000116299-ELAPOR1/pathology/pancreatic+cancer#img; https://www.proteinatlas.org/ENSG00000086619-ERO1B/tissue/pancreas#img; https://www.proteinatlas.org/ENSG00000086619-ERO1B/pathology/pancreatic+cancer#img; https://www.proteinatlas.org/ENSG00000091129-NRCAM/tissue/pancreas#img; https://www.proteinatlas.org/ENSG00000091129-NRCAM/pathology/pancreatic+cancer#img accessed on 24 November 2020). The differences in the GEPIA database were determined using one-way ANOVA, * *p*-value ˂ 0.05.

### 4.9. miRNA Target Gene Identification

The target genes of miRNAs were identified using the miRPathDB V2.0 (https://mpd.bioinf.uni-sb.de/ accessed on 10 November 2020) [62] database, and the target genes with experimental evidence and prediction were selected. Pathways and biological processes were analyzed using the Kyoto Encyclopedia of Genes and Genomes (KEGG) and Gene Ontology (GO) database in the Enrich database [63]. We set a *p*-value ˂ 0.05 as the cutoff criterion. Finally, the *p*-value was −log10 transformed.

### 4.10. ROC Curve Analysis

ROC curve analysis was performed using five different software programs, Graph Prism v8.0.2 (GraphPad Software, La Jolla, CA, USA), Analyses-it V5.66 (https://analyze-it.com/ accessed on 19 November 2020), MedCalc^®^ V19.6 (www.medcalc.org; accessed on 6 December 2022), R Studio v4.2.1 (www.R-project.org accessed on 1 February 2021) [64] and easyROC V1.3.1 statistical analysis software (www.biosoft.hacettepe.edu.tr/easyROC accessed on 3 February 2021). The AUC was calculated, and an AUC ≥ 7 with a confidence interval (CI) of 95% and a *p*-value ˂ 0.05 was considered good [23]. Optimal cutoff, sensitivity, and specificity were determined using the Youden index [65].

### 4.11. ROC Curve Combinations

The expression levels of the four DEmiRNAs (miR-222-3p, miR-345-5p, miR-100-5p, and miR-221-3p) from control and PDAC plasma samples were used to create ROC curves employing Graph Prism software v8.0.2. For each combination, the expression levels of these miRNAs were organized by columns in new, independent tables. Two columns are included in each table: one for controls (A) and the other for the miRNAs of patients with PDAC (B). By comparing the expression levels of the two, three, or four miRNA combinations found in columns A and B, different combinations were generated using these columns. These columns included the expression levels of two (miR-222-3p/miR-345-5p, miR-222-3p/miR-100-5p, miR-222–3p/miR-221-3p, miR-345-55p/miR-100-5p, miR-345-5p/miR-221-3p, miR-100-5p/miR-221-3p), three (miR-222-3p/miR-345-5p/miR- 100-5p, miR-222-3p/miR-345-5p/miR-221-3p, miR-100-5p/miR-221-3p/miR-222-3p, miR-100-5p/miR-221-33p/miR-345-5p) and four miRNA combinations (miR-222-3p/miR-345-5p/miR-100-5p/miR-2221-3p), resulting in the generation of 11 tables, one table for each combination of miRNAs examined. Using the default parameters, ROC curves were generated using these data. Finally, the AUC was calculated for each combination using a 95% CI, and an AUC of less than 7 with a value of 0.05 was considered sufficient. Finally, the AUC was calculated for each combination with a 95% CI, and an AUC ≥ 7 with a value of *p* ˂ 0.05 was considered sufficient [23]. Using the Youden index method, the best cutoff values, sensitivity, and specificity for each miRNA combination were identified [65]. Data on sensitivity and specificity are shown for each combination as percentages.

### 4.12. Survival Analysis

OS was analyzed in patients with PDAC in the TCGA dataset using the Kaplan-Meier Plotter database (http://kmplot.com accessed on 25 November 2020). The median expression was selected as the best cutoff for each miRNA, and the rest of the parameters selected were by default. The patients were classified according to miRNA expression as high (red line) or low (black line). The OS analysis of the miR-222-3p/miR-221-3p combination was performed considering the mean expression of these miRNAs. Hazard ratios (HRs) with 95% CIs and *p*-values were calculated using the logrank test. A *p*-value ˂ 0.05 was considered statistically significant.

### 4.13. Statistical Analysis

All data were analyzed in Graph Prism v8.0.2 software (GraphPad Software, La Jolla, CA, USA). Data are shown as medians with 95% CIs. The differences were analyzed using the Mann-Whitney test, and a *p*-value of ˂0.05 was considered statistically significant.

## 5. Conclusions

We found that miR-221-3p and miR-222-3p have the best PDAC prognostic value, together with their 14 possible target genes. Our findings identified a set of molecules (miRNAs and their target genes) as potential biomarkers of poor PDAC prognosis. Finally and importantly, we identified a new panel consisting of two miRNAs (miR-222-3p and miR-221-3p) that can be considered good noninvasive biomarkers for diagnosing PDAC. Therefore, miR-221-3p and miR-222-3p may be good candidates for developing a noninvasive and specific diagnostic method for managing this devastating cancer.

## Figures and Tables

**Figure 1 ijms-24-15193-f001:**
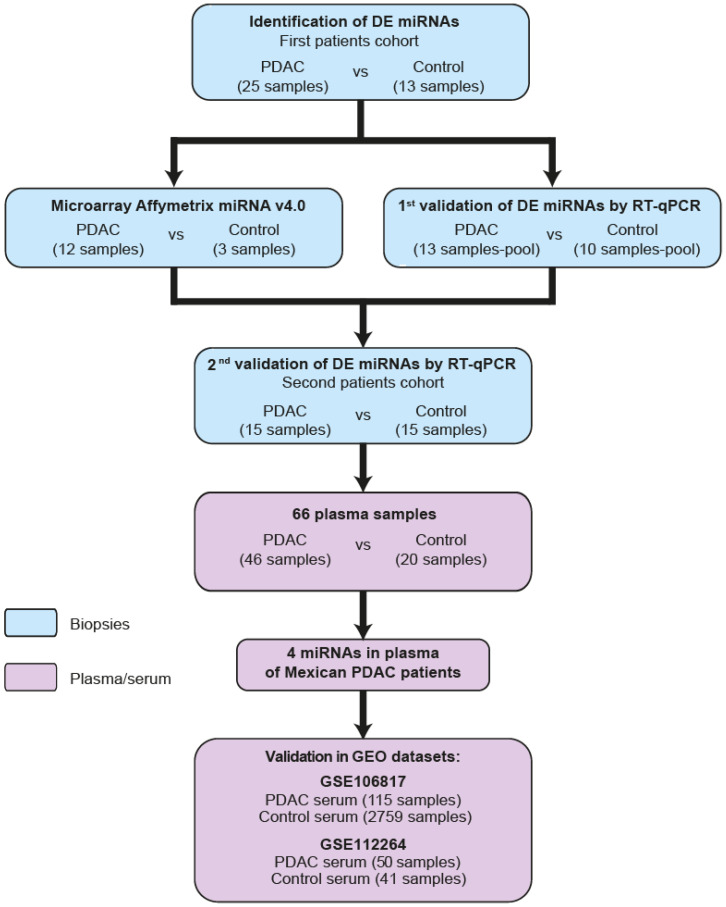
Flow chart integrating study design. Workflow of PDAC patients and control used in this study.

**Figure 2 ijms-24-15193-f002:**
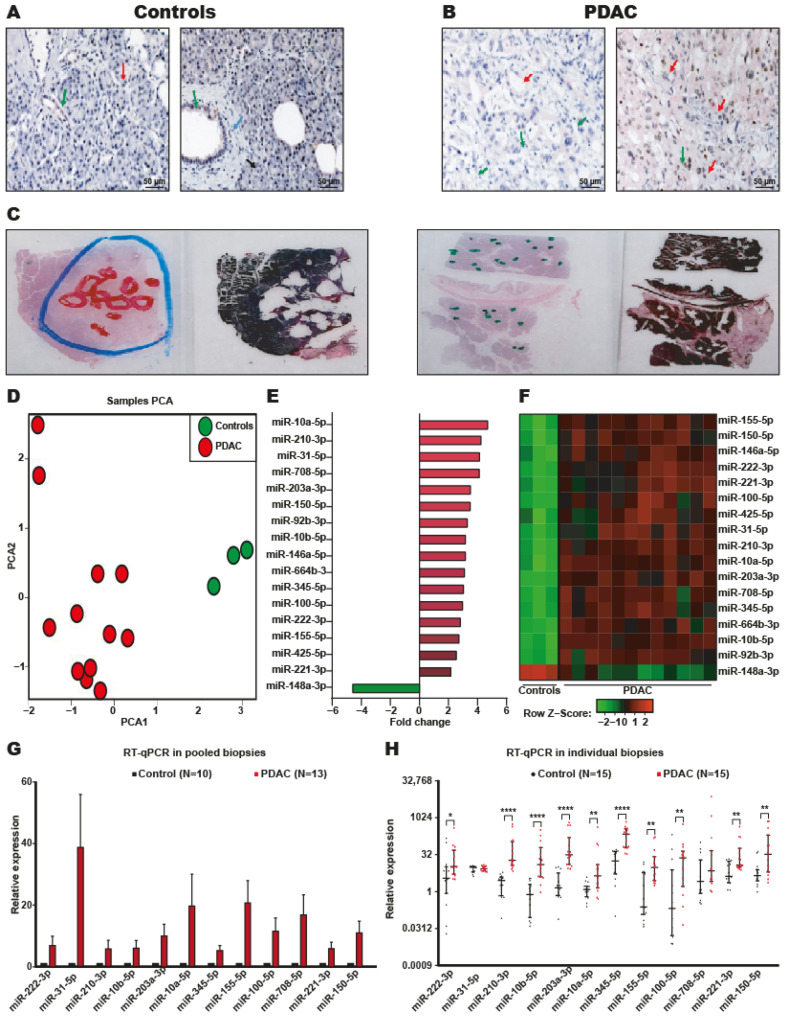
Immunohistochemistry of neoplastic and non-neoplastic pancreatic tissue and ten DEmiRNAs in biopsies from PDAC patients. Normal and PDAC tissues were analyzed by IHC using an anti-CK7 antibody (pink). (**A**) Two representative images of normal pancreatic tissue at 50X are shown. Arrow green corresponds to the duct with its well-differentiated simple cubic epithelium. Acini red arrows. Lobules of regularly arranged acini are indicated by the black arrow. Connective tissue is indicated by blue arrows. Scale bars = 50 μm. (**B**) Images of neoplasia pancreatic tissue at 50X are shown. PDAC with a moderate degree of differentiation (**left**) and PDAC tissue with a low degree of differentiation (**right**). The red arrow shows glandular structures with the proliferation of neoplastic ducts. The green arrow shows desmoplastic stroma. (**C**) Macrodissection of neoplastic (**left**) and nonneoplastic tissue (**right**). The tumor is indicated by a blue circle, while areas with a higher density of neoplastic ductal cells are shown in red circles (**left**). Areas with the highest number of ductal cells are marked with green color in nonneoplastic tissue (**right**). In both cases, the tissue is shown after microdissection. (**D**) PCA was performed on miRNA expression levels in tissue from PDAC patients and healthy subjects. Each red circle corresponds to 12 PDAC biopsies, and the green circle represents three control biopsies. According to the graph, the tumor and control samples were clustered at different localities (Left panel). (**E**) Changes in the expression levels of the 17 DEmiRNAs between PDAC and non-neoplastic pancreatic tissue. The fold change is shown on the *x*-axis. (**F**) Heatmap of the miRNA microarrays using 12 PDAC samples and three pancreatic non-neoplastic samples. Hierarchical grouping of the 16 over-expressed miRNAs (red) and 1 decreased miRNA between PDAC and non-neoplastic pancreatic tissue. (**G**) RT-qPCR assays were performed to validate the top 12 DEmiRNAs in the PDAC biopsies by using pooled biopsies of 10 control samples and 13 PDAC samples that remained from our first cohort of patient microarrays. (**H**) RT-qPCR assays were performed in a second cohort of individual samples from patients with PDAC and from non-neoplastic samples to corroborate the presence of the 12 DEmiRNAs in PDAC. RNU6 was used as the housekeeping gene. The differences were determined using the Mann–Whitney test. * *p*-value ˂ 0.05, ** *p*-value ˂ 0.01 and **** *p*-value ˂ 0.0001.

**Figure 3 ijms-24-15193-f003:**
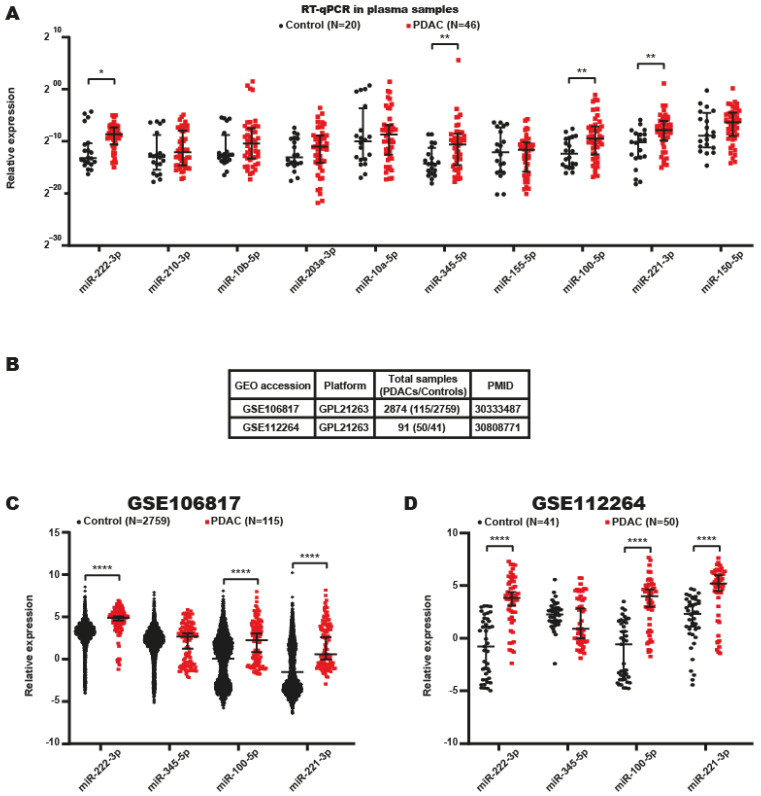
miRNAs found in the plasma of PDAC patients. (**A**) Expression profiles of circulating miRNAs in the plasma of 46 patients with PDAC and 20 healthy controls that live in Mexico. The differences were determined using the Mann–Whitney test. * *p*-value ˂ 0.05 and ** *p*-value ˂ 0.01. (**B**) Table summarizing the information from GSE106817 and GSE112264 used in this study. (**C**,**D**) Analysis of expression of four DEmiRNAs from the GSE106917 and GSE112264 datasets, respectively. The differences were determined using the Mann–Whitney test. **** *p*-value ˂ 0.0001.

**Figure 4 ijms-24-15193-f004:**
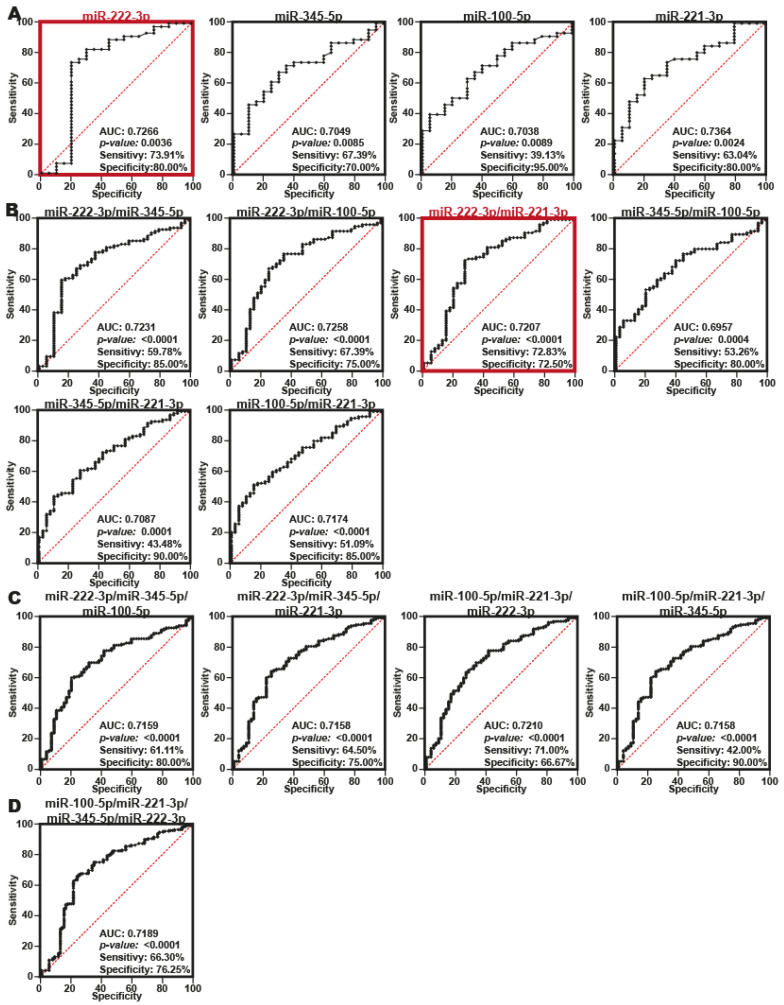
The miRNA signature of miRNA-221-3p and miR-222-3p could be considered promising molecules for the PDAC diagnostic. (**A**) Individual ROC curves of four DEmiRNA signatures to identify PDAC patients vs. control people from individuals that live in Mexico. The graph in red box indicated the most accurate biomarker among the four miRNAs discovered. (**B**–**D**) ROC curves combination of two, three and four miRNAs were used to identify the best miRNAs combination to be considered for the PDAC diagnostic in PDAC patients of the three miRNAs that could be used in the PDAC diagnosis. In (**B**), the graph in red box indicated the most accurate biomarker between miRNAs combinations.

**Figure 5 ijms-24-15193-f005:**
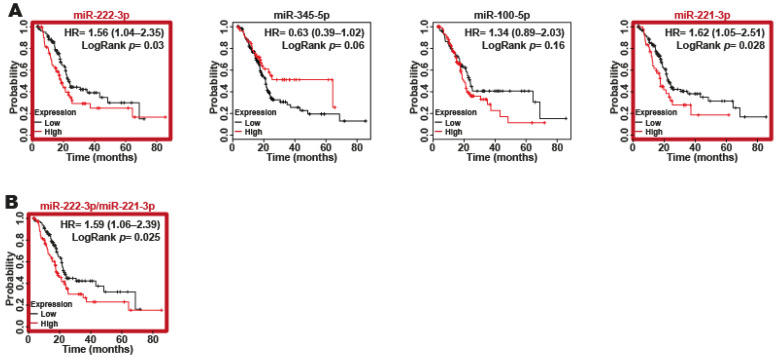
Two of the four DEmiRNAs may be used as biomarkers of poor prognosis in PDAC. (**A**) The association between the individual expression of four DEmiRNAs and the OS rate in patients with PDAC analyzed through Kaplan–Meier curves. (**B**) The combination of miR-222-3p and miR-221-3p expression is the best miRNA combination to determine poor survival in PDAC patients. HR with 95% CI and *p*-value were calculated using the logrank test. A *p*-value ˂ 0.05 was considered significant.

**Figure 6 ijms-24-15193-f006:**
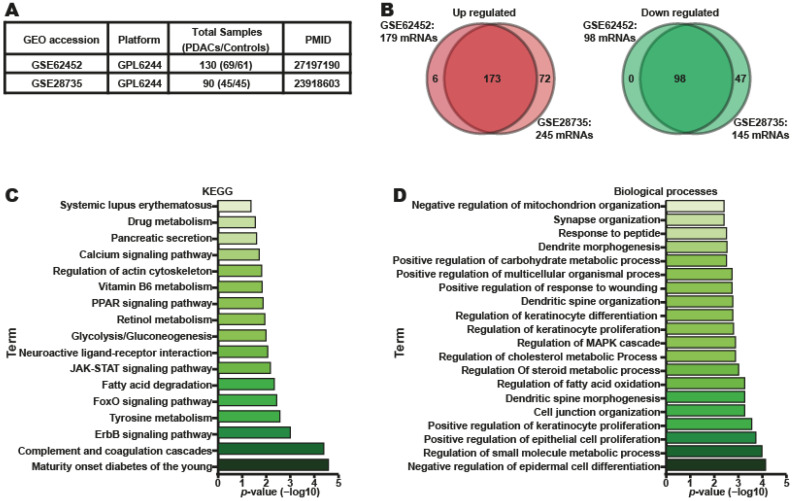
Biological process and KEGG pathway analysis of the mRNAs target from four DEmiRNAs identified in PDAC patients’ plasma. (**A**) Table compiling the data from GSE62452 and GSE28735 used in this research. (**B**) DEGs are found in both datasets. Comparison of the shared and unique mRNAs target of four DEmiRNAs up- and down-regulated. (**C**) Enriched pathways for the four DEmiRNAs’ 47 downregulated targets that were identified by KEGG pathway analysis. (**D**) The enriched GO biological processes in which 47 potential mRNA targets from four DEmiRNAs are involved. A *p*-value < 0.05 was considered statistically significant.

**Table 1 ijms-24-15193-t001:** Clinico-pathologic features of samples used in this work.

Type of Sample	Characteristic	Controls n = 48	Patients n = 86
	Age		
Tissue (n = 68)	Median (range)	50 (26–78)	63 (40–86)

Plasma (n = 66)	Median (range)	46 (27–62)	63 (41–83)

	Gender		
Tissue (n= 68)	Male	17	17
	Female	11	23

Plasma (n = 66)	Male	9	21
	Female	11	25

	Alcohol		
Tissue (n = 68)	Yes	0	0
	No	0	0
	No data	28	40

Plasma (n = 66)	Yes	0	9
	No	0	31
	No data	20	6

	Smoking		
Tissue (n = 68)	Yes	0	0
	No	0	0
	No data	28	40

Plasma (n = 66)	Yes	0	16
	No	0	20
	No data	20	10

	Ethnicity		
Tissue (n = 68)	Mexican-Mestizo	28	40

Plasma (n = 66)	Mexican-Mestizo	20	46

	Tumor size		
Tissue (n = 68)	≤4 cm	-	9
	>4 cm	-	18
	No data	-	13

Plasma (n = 66)	≤4 cm	-	1
	>4 cm	-	0
	No data	-	45

	TNM		
Tissue (n = 68)	I	-	0
	II	-	18
	III	-	8
	IV	-	11
	No data	-	3

Plasma (n = 66)	I	-	1
	II	-	5
	III	-	18
	IV	-	20
	No data	-	2

	Differentiation grade		
Tissue (n = 68)	G1	-	2
	G2	-	30
	G3	-	7
	No data	-	1

Plasma (n = 66)	G1	-	0
	G2	-	0
	G3	-	0
	No data	-	46

	Survival (Tissue/Plasma)		
Tissue (n = 68)	Short: ≤14 months	-	15
	Long: >14 months	-	7
	No data	-	18

Plasma (n = 66)	Short: ≤14 months	-	0
	Long: >14 months	-	0
	No data	-	46

-: Is not applicable.

**Table 2 ijms-24-15193-t002:** Seventeen miRNAs were differentially expressed in the tissue of PDAC patients.

Transcript_ID	logFC	AveExpr	t	*p*-Value	FDR
Overexpressed miRNAs					
miR-222-3p	2.8844	10.8590	10.7525	9.96 × 10^−9^	6.41 × 10^−7^
miR-31-5p	4.1967	9.7829	10.7844	9.56 × 10^−9^	6.53 × 10^−7^
miR-210-3p	4.3097	8.0389	10.3553	1.69 × 10^−8^	7.75 × 10^−7^
miR-10b-5p	3.2291	6.3356	10.5711	1.27 × 10^−8^	9.64 × 10^−7^
miR-203a	3.5790	6.0323	9.9259	3.06 × 10^−8^	1.30 × 10^−6^
miR-10a-5p	4.4917	8.5372	9.5532	5.19 × 10^−8^	2.38 × 10^−6^
miR-345-5p	3.0996	5.9820	8.6866	1.88 × 10^−7^	2.88 × 10^−6^
miR-155-5p	2.7886	9.5953	8.4357	2.78 × 10^−7^	3.96 × 10^−6^
miR-100-5p	3.0294	10.4069	8.2603	3.66 × 10^−7^	4.38 × 10^−6^
miR-708-5p	4.1888	6.7173	8.0084	5.49 × 10^−7^	6.25 × 10^−6^
miR-221-3p	2.2323	10.7475	7.6056	1.06 × 10^−6^	1.10 × 10^−5^
miR-146a-5p *	3.2269	8.8496	7.5014	1.27 × 10^−6^	1.26 × 10^−5^
miR-150-5p	3.5555	8.7285	7.4770	1.32 × 10^−6^	1.27 × 10^−5^
miR-664b-3p	3.1629	5.9450	8.1304	4.51 × 10^−7^	1.35 × 10^−5^
miR-92b-3p	3.3605	5.8241	7.3678	1.59 × 10^−6^	1.45 × 10^−5^
miR-425-5p	2.6038	8.3962	6.4191	8.51 × 10^−6^	7.36 × 10^−5^
Downregulated miRNA					
miR-148a-3p	−4.5795	7.7280	−9.7445	3.95 × 10^−8^	9.37 × 10^−7^

* We cannot amplify this miRNA by PCR.

**Table 3 ijms-24-15193-t003:** Primer sequences used to amplify each miRNA identified in this study.

miRNA	Forward	Reverse
miR-222-3p	GCAGAGCTACATCTGGCT	CCAGTTTTTTTTTTTTTTTACCCAGT
miR-31-5p	GCGCAGCTGTGCGTGTGACA	GTCCAGTTTTTTTTTTTTTTTAGCTATG
miR-210-3p	GCGCAGCTGTGCGTGTGACA	GTTTTTTTTTTTTTTTCAGCCGCT
miR-10b-5p	CAGTACCCTGTAGAACCGA	GGTCCAGTTTTTTTTTTTTTTTCAC
miR-203a	CAGGTGAAATGTTTAGGACCA	GGTCCAGTTTTTTTTTTTTTTTCTAGT
miR-10a-5p	GCAGTACCCTGTAGATCCGA	GGTCCAGTTTTTTTTTTTTTTTCAC
miR-345-5p	GGCTGACTCCTAGTCCAG	GGTCCAGTTTTTTTTTTTTTTTGAG
miR-155-5p	CGCAGTTAATGCTAATCGTGATAG	GGTCCAGTTTTTTTTTTTTTTTAACC
miR-100-5p	CAGAACCCGTAGATCCGA	GTCCAGTTTTTTTTTTTTTTTACAAG
miR-708-5p	CAGAAGGAGCTTACAATCTAGC	GTCCAGTTTTTTTTTTTTTTTCCCA
miR-221-3p	GCAGAGCTACATTGTCTGCT	CAGTTTTTTTTTTTTTTTGAAACCCA
miR-150-5p	AGTCTCCCAACCCTTGTACCA	GGTCCAGTTTTTTTTTTTTTTTCACT
RT-primer	CAGGTCCAGTTTTTTTTTTTTTTTVN	
RNU6	CTCGCTTCGGCAGCACATATACT	ACGCTTCACGAATTTGCGTGTC
miR-39-3p	GTCACCGGGTGTAAATCAG	GGTCCAGTTTTTTTTTTTTTTTTTCAAG

## Data Availability

The data discussed in this publication have been deposited in NCBI’s Gene Expression Omnibus and are accessible through GEO accession number GSE163031.

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
