# Peer review of "Circulating miRNAs as Noninvasive Biomarkers for PDAC Diagnosis and Prognosis in Mexico"

_ijms, 2023, doi:10.3390/ijms242015193_

Round 1
Reviewer 1 Report
Language needs improvement
Language needs improvement
Author Response
Q1Language needs improvement
A1. The English edition of the document was done by Journal Experts. Please see, the attached file the supporting document.
Reviewer 2 Report
Alvarez-Hilario and colleagues investigated the miRNA profile of PDAC tumors and healthy control tissues of Mexicans and identified miRNAs that are dysregulated using microarray. Subsequently, they validated their genes in a tissue samples followed by plasma samples. Finally, the author evaluated the downstream genes using in silico dataset. The study design is not novel (similar to that of previous studies) and the sample size appears to be relatively small. There are several analyses in the paper which appears to not to add to the paper and highly speculative, those analyses should be deleted from the manuscript. Considering that the authors highlight the use of Mexican cohort, it would be valuable if the authors focused more on the novelty of the samples and how this cohort, differs from other published clinical cohorts.
Specific comments
DEmiRNA seems misleading. Use “DE miRNA”, since it is not a subtype of RNA and rather it is a type of analysis.
Figure 2D. Please make your data display consistent for Figure 2D (use the same format as 2E).
While the authors have provided individual subject data, it would be beneficial to include a summary table that facilitates comparisons between the different groups. Provide a table which offer a concise overview of the clinical characteristics of the subjects in each group used in the analysis, aiding in the interpretation of the study's findings.
Considering that this study utilized a unique subject cohort, additional information on the subjects used in the study would be of use. Please provide race/ethnicity information or any other additional clinical information that would add to the paper.
Figure 3A. Please provide the individual dots for controls. It would be great for the readers to see the variabilities for controls.
It is not clear that how multiple miRNAs are combined. Please provide more detailed methods in the method section as well as the results.
It is not clear that why the prognostic potential of these miRNA candidates was tested. Dysregulation of miRNA does not necessarily result in prognosis. Please provide a reasonable explanation for performing prognostic analysis and why do you think that genes that you’ve selected are legitimately important.
Instead of using mRNA sequencing data in silico, please use in silico miRNA data to validate your data. Both tissue and plasma/serum data if available.
Can you validate the prognostic potential of miRNAs that you’ve identified in your own cohort?
Minor English editing should improve overall readability of the paper.
Reviewer 3 Report
In this manuscript “miRNAs in Circulation as non-invasive biomarkers for the Diagnosis and Prognosis of PDAC in Mexico” authors presented an exploration of miRNAs in PDAC samples from Mexican cohort. The validation was further performed on an independent internal as well as 2 external cohorts. The mapping of 2 miRNAs in independent external data sets which were further utilized for downstream analysis highlight the importance of these miRNAs as an important biomarker for PDAC in Mexican cohort. This study specifically in Mexican population further support the validation of these miRNAs as biomarkers, although they have been reported in previous PDAC studies.
Author Response
.

Reviewer 4 Report
In the manuscript: miRNAs in Circulation as non-invasive biomarkers for the 2 Diagnosis and Prognosis of PDAC in Mexico The authors found fourth DEmiRNAs in the mi-35 croarray tissues and plasma from PDAC patients. Two Gene Expression Omnibus datasets included serum samples from PDAC patients validated their overexpression. They discovered a brand-new panel made up of two miRNAs (miR-222-3p and miR-221-3p) that can be used as reliable noninvasive biomarkers to diagnose PDAC. The survival rates of PDAC patients decreased when miR-221-3p and miR-222-3p were overexpressed in 39. In conclusion, miR-222-3p and miR-221-3p may be effective noninvasive biomarkers for the diagnosis of PDAC in persons who live in Mexico, as well as miR-221-3p and miR-223-3p for the prognosis of the disease. The authors concentrated on a number of traits that make a biomarker beneficial for diagnosing PDAC, including the p-value, sensitivity, specificity, and AUC. The results demonstrate that, when all of these factors were taken into account, the miR-222-3p/miR-221-3p combination displayed a higher mix of specificity, sensitivity, and p-value.The paper deals with an interesting and translational scientific topic
Despite the excellent writing and organization of the report, I have some suggestions:
1) In line 94-96, the authors investigated the presence of miR-222-3p/miR-221-3p in a cohort of PDAC and normal tissue. However, they provided only minimal information regarding the samples' provenance, age, and gender size of the histological lesion,. I would suggest adding a summary table
2) The authors discuss the histological differences between healthy pancreatic tissue and PDAC, and may include figure S1A-B in the manuscript?
3) Is there evidence of the same miRNAs in other forms of cancer?
4) Is the analysis in plasma suggesting the alteration of only 4 miRNAs quantitative? Could a correlation be established between the amount of miRNA in plasma and tumor grading?

Minor English editing required
Author Response
.

Reviewer 5 Report
The article "miRNAs in Circulation as non-invasive biomarkers for the Diagnosis and Prognosis of PDAC in Mexico" is an interesting scientific study on the possibility of early diagnosis of pancreatic cancer. The article begins with a concise abstract and introduction. It is also important that the Authors include a methodological abstract. The Materials and methods chapter is detailed. Although the microarray technique has been used for quite a long time, it is a very sensitive and specific method used in genetic research. The items in the references section are up-to-date, and the research results contained therein were mostly developed in the last 10 years. Due to the scientific and prognostic value of the presented research results, I propose to accept the article "miRNAs in Circulation as non-invasive biomarkers for the Diagnosis and Prognosis of PDAC in Mexico" for publication in IJMS in its present form.
A minor edition of the English language is required.
Author Response
.

Round 2
Reviewer 2 Report
Authors have sufficiently addressed all the comments.